

# The inter-trial and inter-session reliability of Theia3D-derived markerless gait analysis in tight versus loose clothing

Sylvia Augustine[1], Richard Foster[1], Gabor Barton[1], Mark J. Lake[1], Raihana Sharir[1,2] and Mark A. Robinson[1]

[1] School of Sport and Exercise Sciences, Liverpool John Moores University, Liverpool, United Kingdom
[2] Faculty of Sports Science and Recreation, Universiti Teknologi MARA (UiTM), Shah Alam, Selangor, Malaysia

## ABSTRACT

**Background.** Gait analysis is traditionally conducted using marker-based methods yet markerless motion capture is emerging as an alternative. Initial studies have begun to evaluate the reliability of markerless motion capture yet the evaluation of different clothing conditions across sessions and complete evaluation of the lower limb and pelvis reliability have yet to be considered. The aim of this study was to evaluate the inter-trial, inter-session and inter-session-clothing variation and root mean square differences between tight- or loose-fitting clothing during walking.

**Method.** Twenty-two healthy adult participants walked along an indoor walkway whilst eight video cameras recorded their gait in either tight- or loose-fitting clothing. A commercial markerless motion capture system (Theia3D) provided gait kinematics for evaluation.

**Results.** Reliability results showed average inter-trial variation of <2°, inter-session variation of <3° and inter-session-clothing variation <3.5°. Root mean square differences (RMSD) between clothing conditions were <2°.

**Discussion.** Pelvis variations were smaller than those at the hip, knee and ankle. Our results showed smaller variation than in previous studies which may be due to updates to software. The demonstration of the reliability of markerless motion capture for gait analysis in healthy adults should prompt further evaluation in clinical conditions and reconsideration of multi-assessor marker-based gait analysis protocols, where variation is highest.

Corresponding authors
Sylvia Augustine,
s.b.augustine@2021.ljmu.ac.uk
Mark A. Robinson,
m.a.robinson@ljmu.ac.uk

## INTRODUCTION

Gait analysis is a common procedure used in both clinical and research settings. A typical gait analysis examines kinematics at the pelvis, hip, knee, and ankle in one complete gait cycle. Kinematics are routinely obtained by sticking reflective markers on to anatomical locations of participants which requires a trained tester. The position of these markers is captured by infra-red motion capture cameras in a laboratory setting to establish the kinematics of the participant as they walk. It is essential for participants to wear minimal

or tight-fitting clothing to allow careful placement of markers for accurate kinematic data (*Riazati et al., 2022*) and to minimise variation (*Gorton III, Hebert & Gannotti, 2009*).

Variation in marker-based gait analysis can be attributed to different sources of experimental error (*Schwartz, Trost & Wervey, 2004*). Intrinsic variation (or inter-trial variation) describes the natural repeatability of gait patterns including fluctuations in walking speed, whereas extrinsic variation (or inter-session variation) adds the effects of repeating the experimental process on to the inter-trial variation. Variation in marker-based gait measurement have been comprehensively examined with differing variation seen between joints, models and planes (see *McGinley et al., 2009* for a review). *McGinley et al. (2009)* synthesised summary provided the suggestions that gait kinematic variation <2° might be "acceptable", variation between 2–5° might be "reasonable" and >5° degrees may "raise concern". The Clinical Movement Analysis Society UK and Ireland CMAS, (*Stewart et al., 2023*) also provide guideline thresholds of 5° for intra-tester repeatability. Although these are global thresholds, that may not be universally be accepted, they are a useful benchmark to evaluate new technologies against.

Markerless motion capture is an attractive alternative to marker-based motion capture because it can also generate kinematics whilst potentially mitigating some of the errors with placing markers (*Colyer et al., 2018*). Markerless motion capture hardware/software systems range from single to multi-camera approaches (*Wade et al., 2022*) *e.g.*, Microsoft Kinect, Captury, SIMI Shape 3D motion capture, OpenPose, DeepLabCut, Theia3D, OpenCap, each has its own merits and limitations. This current study specifically considers Theia3D (Theia Markerless Inc., Kingston, ON, Canada) as a leading commercial system that was available to us. Theia3D uses deep-learning-algorithms to recognize important anatomical features from multiple 2D videos to create a three-dimensional (3D) inverse kinematic pose of body segments (*Kanko et al., 2021c*). As segmental kinematics are determined algorithmically, markers are not physically placed on participants, and, therefore, markerless motion capture may not be susceptible to the same errors as marker-based motion capture. This also means that participants may not need to wear tight-fitting clothing for repeatable kinematics.

Initial studies have begun to evaluate the reliability of Theia3D for gait analysis, with specific consideration of spatiotemporal parameters (*Kanko et al., 2021c*; *McGuirk et al., 2022*; *Riazati et al., 2022*), non-pathological gait (*Kanko et al., 2021b*; *Riazati et al., 2022*) different clothing types (*Keller et al., 2022*) and pathological gait (*McGuirk et al., 2022*; *Outerleys et al., 2024*). Results to date show that the average inter-trial variation across planes and joints appear to be within reasonable limits at 2.5° (*McGinley et al., 2009*) with inter-session variation increasing average variation to 2.8° (*Kanko et al., 2021a*) and 2.85° (*Riazati et al., 2022*). Averages though can mask higher variation in particular joints, planes and times, as transverse plane results had higher variation than other planes (*Kanko et al., 2021a*; *Riazati et al., 2022*). Inter-session variation of a markerless system remains pertinent to consider as factors such as calibration, lighting, camera positioning, background contrast and participant clothing may all reasonably change in different laboratories and between sessions. What is notably absent in the above studies, however, is consideration of the repeatability of the pelvis kinematics. Pelvis kinematics are typically reported in all planes

during a gait analysis to inform clinical decision making and are 3/9 angle inputs of the Gait Deviation Index and Movement Deviation Profile, which are example gait indices that summarise how gait differs from a normative cohort (*Schwartz & Rozumalski, 2008*; *Barton et al., 2012*). The reliability of pelvis kinematics should therefore also be determined.

The initial markerless reliability studies using Theia3D (*Kanko et al., 2021a*; *McGuirk et al., 2022*; *Riazati et al., 2022*) deliberately made no attempt to control the clothing worn by participants. As Theia3D identifies anatomical features, different clothing may affect landmark identification, and, therefore, joint kinematics. We are only aware of one study (*Keller et al., 2022*) to date that has evaluated the effect of clothing on the reliability of gait kinematics. In their comparison of "sport" and "street" clothing, the average RMSD of 2.6° across all joints and planes was observed, and a range of 1.4° (frontal plane hip) to 4.2° (ankle in-out toeing) reported. The consideration of different clothing types is relevant because gait analysis participants are often required to change from their preferred clothes into tight-fitting clothes for marker-based gait analysis. With further evaluation of the effects of clothing type we can begin to establish whether tight-fitting clothes are beneficial to the reliability of markerless gait analysis. *Keller et al. (2022)* clothing evaluation occurred within the same session, yet a likely gait analysis situation is also where participants return on a separate day in different clothing. The variation associated with a combination of a second session and altered clothing, what we subsequently call the inter- session-clothing variation, should, therefore, be quantified.

This study aims to (1) evaluate the inter-trial and inter-session gait variation including pelvis kinematics, (2) evaluate the effect of clothing within and between sessions.

## MATERIALS & METHODS

### Participants

A convenience sample of 22 healthy participants (8 = female, 14 = male, mean $\pm$ SD age: 25 $\pm$ 4 years; height 174 $\pm$ 9 cm, weight 70 $\pm$ 11, body-mass-index: 22 $\pm$ 5 kg/m$^2$) volunteered for this study. Participants were free from lower-limb injury, not undergoing rehabilitation and without neuromuscular or musculoskeletal impairment. All participants provided written informed consent and the study was granted Liverpool John Moores University Research Ethics Committee approval (UREC reference: 21/SPS/063).

### Protocol

Participants attended the biomechanics laboratory for two gait data capture sessions, with the second session within 14 days of the first (mean $\pm$ SD; 6 $\pm$ 3 days). For each session participants completed ten separate overground walking trials at a self-selected pace through a 6 m long $\times$ 3 m wide calibrated volume. Participants brought two sets of clothing for each session. One set was "tight" *e.g.*, a base layer or tight-fitting top exposing the elbow and tight shorts above the knee that would follow body movements. Participants were advised to wear contrasting colours between the top and shorts to avoid participants wearing only one colour of clothing. The second set was "loose", which was loose-fitting regular daily wear without constraints. Participants were required to wear the same pair of shoes for both sessions (Fig. 1) but were not required to bring the same clothing. Five
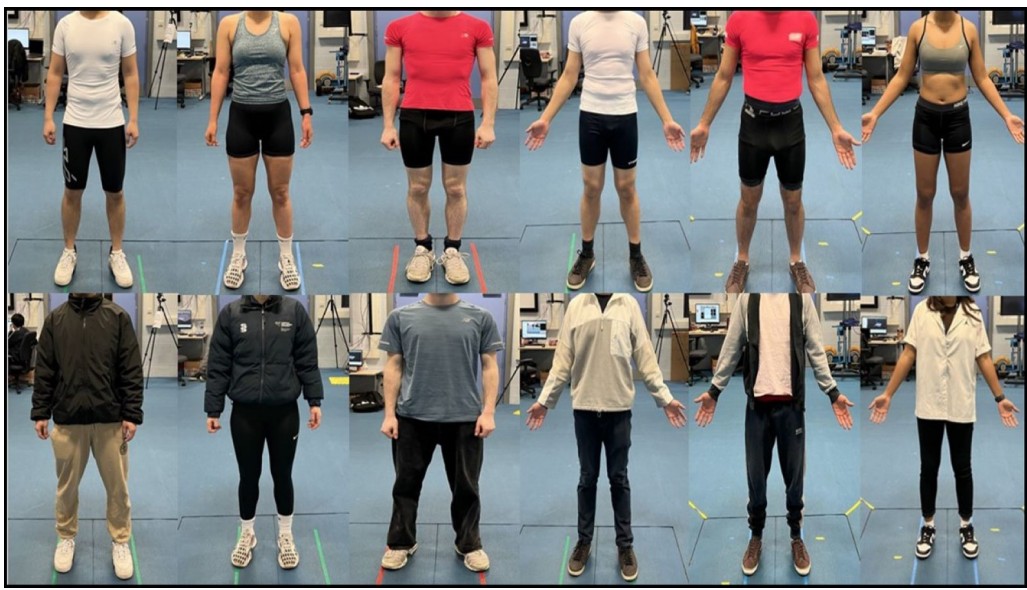

**Figure 1** Example tight- and loose-fitting clothing for six participants.

trials in each clothing condition were completed. Video data were captured using eight synchronized Qualisys Miqus video cameras (Qualisys AB, Gothenburg, Sweden) with a resolution of $1,280 \times 720$ pixels and capture rate of 180 Hz. Cameras were positioned on tripods symmetrically and evenly distributed around the walking volume between 1.2–1.65 m high. Cameras were positioned closer to the volume centre than what would be typical for a 3D camera system to ensure that the participant was at least 500 pixels tall within the calibrated volume.

## Data analysis

Video data were processed using Theia3D software (Theia Markerless Inc., Kingston, ON, Canada, v2022.1.0.2309), which generated their default full-body model. The default model has a six degrees-of-freedom at the pelvis and three rotational degrees-of-freedom at the hip, knee and ankle (*Kanko et al., 2021a*). Theia3D, by default scales a separate inverse kinematic (IK) model to each processed trial, meaning it has an independent model for each clothing condition (*Outerleys et al., 2024*). Model pose estimation generates $4 \times 4$ pose matrices for each segment for all frames using inverse kinematics, smoothed at 10 Hz with a low-pass GVCSPL filter. The filtered Theia3D data was saved as a c3d file and imported to Visual 3D (v.5.02.30, C-Motion, Germantown, MD, USA) for further kinematic analysis.

The gait cycle events toe-off and heel-strike were defined as the maximal displacement of the heel and toe from the sacrum (*Zeni, Richards & Higginson, 2008*). Gait kinematics for each trial, session and clothing condition were calculated in Visual 3D, extracted between right foot heel strike to the next right foot heel strike, then linearly normalized to 101 points and exported to MATLAB2022a (The MathWorks, Natick, MA, USA) for further calculations. Although multiple gait cycles occurred within the capture volume, only the

most central right gait cycle was used for analysis. This was to ensure that the gait cycles analysed did not include the participant initiating their walk and so that the participant would be a similar number of pixels in each camera. A single right-sided gait cycle was chosen so as not to inflate trial numbers. Furthermore separate walking trials were collected to ensure that inter-trial variation represented completely separate walking trials, as would typically be collected during a gait analysis, rather than collecting multiple strides within the same walking trial where one stride might influence a subsequent stride. Intra-trial variation is therefore not considered in this study.

Inter-trial, inter-session and inter-session-clothing variation were calculated across the gait cycle (*Schwartz, Trost & Wervey, 2004*). We utilize the terminology of *Kanko et al. (2021a)* referring to ''variation'' rather than ''error'' with inter-trial variation representing intrinsic variation and inter-session and inter-session-clothing variation representing extrinsic variation, the combination of which contribute to total variability. Inter-trial variation was calculated for each clothing condition from the five trials collected during the first test session. Inter-session variation was calculated for each clothing condition using the five trials collected during the first and second sessions. Inter-session-clothing variation was calculated from all sessions and both clothing conditions (Fig. 2). Variability ratios were calculated as the inter-trial variation divided by the inter-session variation. Summary metrics (averages, maxima, minima and ranges) were calculated from the inter-trial and inter-session variation. The average Root Mean Square Difference (RMSD) was also calculated to describe the average absolute difference between clothing conditions.

Segment lengths for the tibia and femur were also calculated for each trial as the vector distance between the proximal and distal end positions of these segments in Visual3D. The segment lengths were constant throughout a trial so this single value was extracted. Segment lengths were averaged over the five trials of each session and over all participants for each clothing condition. The absolute inter-session difference (in mm) and the difference as a percentage of the session 1 segment length were calculated.

## RESULTS

Minimal inter-trial or inter-session variation differences were observed between clothing conditions (Fig. 3). Inter-trial variation (one day, same clothes), when averaged across stance, joints and planes, were very similar between clothing conditions; loose clothing (1.35 ± 0.43°) had a slightly greater variation than tight clothing (1.29 ± 0.48°). Considering single planes and joints, the average inter-trial variation was <2° (Table 1). Peak inter-trial variation from single instances in the gait cycle reached magnitudes of 3.84° and 2.86° in the sagittal plane for the knee and ankle respectively (Table 2).

Inter-session variation (different days, different clothes, same clothes type), when averaged across stance, joints and planes, were also similar between clothing conditions; loose clothing (2.00 ± 0.56°) again had a slightly greater variation than tight clothing (1.88 ± 0.65°). Notably a second session increased variation by <1°. Considering single planes and joints, average inter-session variation was <3° (Table 1). Peak inter-session variation from single instances in the gait cycle reached magnitudes of 3.77° and 3.57° in the sagittal plane for the knee and ankle, respectively (Table 2). Average variation ratios were 1.51

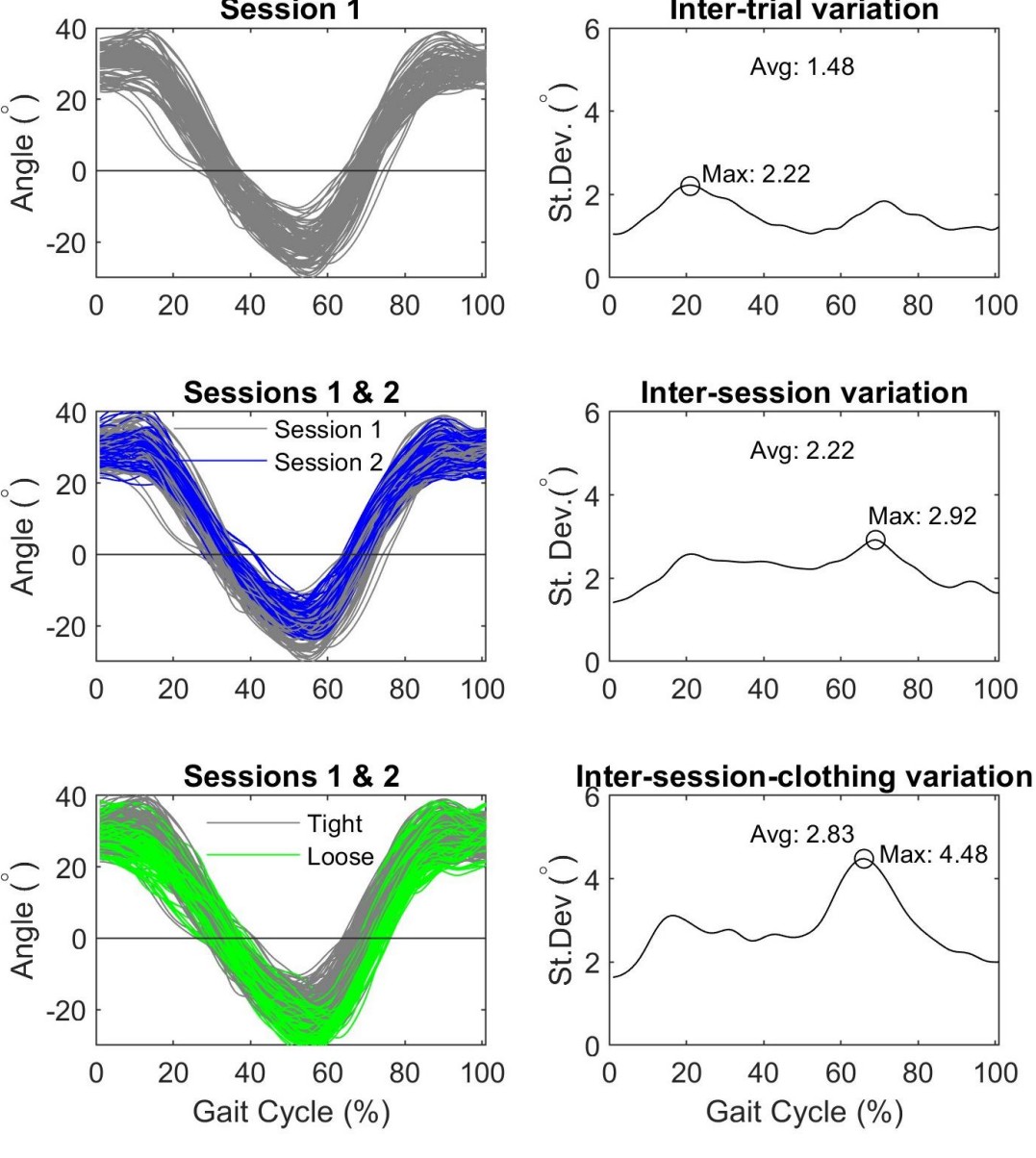

**Figure 2** Example sagittal plane hip angles to illustrate the calculation of inter-trial, inter-session and inter-session-clothing variation and summary metrics (St. Dev. = standard deviation, avg = average, max = maxima).

for loose clothing and 1.49 for tight clothing indicating a second session increased total variation by around 50% (Table 3).

Inter-session-clothing variation (different days, clothes, clothes type) averaged across gait cycle, joints and planes were 2.35 ± 0.61°. Considering single planes and joints, the knee (3.41°) and hip (2.98°) transverse planes had the highest average variation. The peak inter-session-clothing variation occurred at the hip (4.48°) and knee (3.82°) in the sagittal plane.

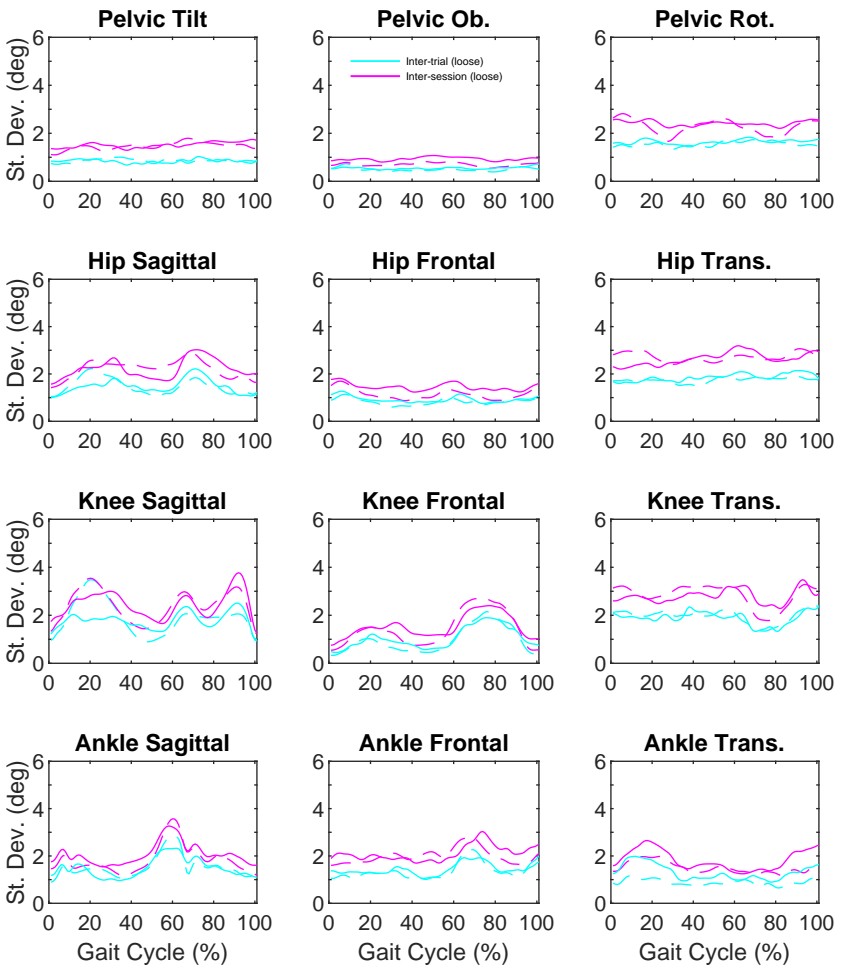

**Figure 3  Inter-trial and inter-session variation in sagittal, frontal and transverse plane lower-limb and pelvis kinematics.** Solid lines represent loose clothing, dashed lines represent tight clothing.

Root mean square differences between the clothing conditions were <2° across all joints and planes (Fig. 4). The highest RMSD was reported at 1.91°, which was hip flexion/extension and the lowest RMSD was 0.89° for pelvic anterior/posterior tilt.

Some differences in segments length between sessions and between clothing conditions were observed (Table 4). In loose clothing the changes in segment lengths between sessions were similar at 2% segment length. In tight clothing the changes in segment lengths between sessions differed between the shank (1%) and thigh (7%).

## DISCUSSION

The aims of this study were to evaluate markerless inter-trial and inter-session gait variation including pelvis kinematics and to evaluate the effect of clothing within and between sessions. Average inter-trial variation was <2°, inter-session variation was <3°,

**Table 1** Average inter-trial, inter-session and inter-session-clothing variation.

| | Sagittal | | Frontal | | Transverse | |
|---|---|---|---|---|---|---|
| | **Loose** | **Tight** | **Loose** | **Tight** | **Loose** | **Tight** |
| Average inter-trial variation (°) | | | | | | |
| Pelvis | 0.84 | 0.85 | 0.55 | 0.50 | 1.64 | 1.55 |
| Hip | 1.48 | 1.48 | 0.92 | 0.83 | 1.86 | 1.74 |
| Knee | 1.79 | 1.90 | 1.04 | 0.97 | 1.89 | 1.96 |
| Ankle | 1.49 | 1.52 | 1.44 | 1.35 | 1.31 | 0.88 |
| Average inter-session variation (°) | | | | | | |
| Pelvis | 1.52 | 1.47 | 0.93 | 0.70 | 2.41 | 2.27 |
| Hip | 2.22 | 2.22 | 1.45 | 1.14 | 2.65 | 2.72 |
| Knee | 2.45 | 2.40 | 1.53 | 1.44 | 2.80 | 2.84 |
| Ankle | 2.07 | 1.84 | 2.11 | 2.00 | 1.83 | 1.50 |
| Avg. inter-session-clothing variation(°) | | | | | | |
| Pelvis | 1.88 | | 1.36 | | 2.72 | |
| Hip | 2.83 | | 1.77 | | 2.98 | |
| Knee | 2.82 | | 1.75 | | 3.41 | |
| Ankle | 2.26 | | 2.45 | | 1.98 | |

**Table 2** The range (Min-Max) for inter-trial, inter-session and inter-session-clothing variation.

| | Sagittal | | Frontal | | Transverse | |
|---|---|---|---|---|---|---|
| | **Loose** | **Tight** | **Loose** | **Tight** | **Loose** | **Tight** |
| Range of inter-trial variation (°) | | | | | | |
| Pelvis | 0.71–1.00 | 0.68–1.03 | 0.39–0.73 | 0.47–0.71 | 1.32–1.73 | 1.43–1.84 |
| Hip | 1.04–2.23 | 1.00–2.22 | 0.60–1.14 | 0.75–1.27 | 1.46–1.89 | 1.59–2.14 |
| Knee | 0.89–3.84 | 0.92–2.51 | 0.33–2.15 | 0.45–1.90 | 1.34–2.39 | 1.32–2.38 |
| Ankle | 0.89–2.86 | 0.96–2.33 | 1.02–2.27 | 1.01–2.01 | 0.65–1.16 | 0.89–1.98 |
| Range of inter-session variation (°) | | | | | | |
| Pelvis | 1.29–1.74 | 1.10–1.79 | 0.80–1.07 | 0.54–0.82 | 2.21–2.61 | 1.66–2.82 |
| Hip | 1.57–3.02 | 1.42–2.92 | 1.23–1.81 | 0.87–1.69 | 2.21–3.19 | 2.41–3.07 |
| Knee | 1.26–3.77 | 1.25–3.54 | 0.75–2.40 | 0.54–2.70 | 2.28–3.48 | 1.78–3.28 |
| Ankle | 1.54–3.27 | 1.17–3.57 | 1.70–3.03 | 1.60–2.80 | 1.25–2.65 | 1.18–1.98 |
| Range of inter-session-clothing variation (°) | | | | | | |
| Pelvis | 1.48–2.39 | | 1.08–1.92 | | 2.06–3.36 | |
| Hip | 1.63–4.48 | | 1.47–2.20 | | 2.76–3.26 | |
| Knee | 1.72–3.82 | | 0.80–2.83 | | 2.92–3.77 | |
| Ankle | 1.68–3.65 | | 2.01–3.24 | | 1.47–2.94 | |

inter-session-clothing variation <3.5°, RMSD between clothing conditions were <2° and inter-session segment length differences ranged from 1–7%.

Inter-trial and inter-session variation in this study were lower than reported in other markerless studies (*Kanko et al., 2021a*; *Keller et al., 2022*) and similar to marker-based studies (*Schwartz, Trost & Wervey, 2004*; *Manca et al., 2010*; *Kaufman et al., 2016*). Inter-session variation with the same clothing added an average of 1° variations. Variability

**Table 3  Average variation ratios for clothing conditions and joints.**

| | Sagittal | | Frontal | | Transverse | |
|---|---|---|---|---|---|---|
| | **Loose** | **Tight** | **Loose** | **Tight** | **Loose** | **Tight** |
| Pelvis | 1.81 | 1.73 | 1.72 | 1.43 | 1.47 | 1.48 |
| Hip | 1.51 | 1.54 | 1.59 | 1.38 | 1.42 | 1.57 |
| Knee | 1.37 | 1.33 | 1.54 | 1.54 | 1.5 | 1.45 |
| Ankle | 1.41 | 1.21 | 1.48 | 1.52 | 1.41 | 1.71 |

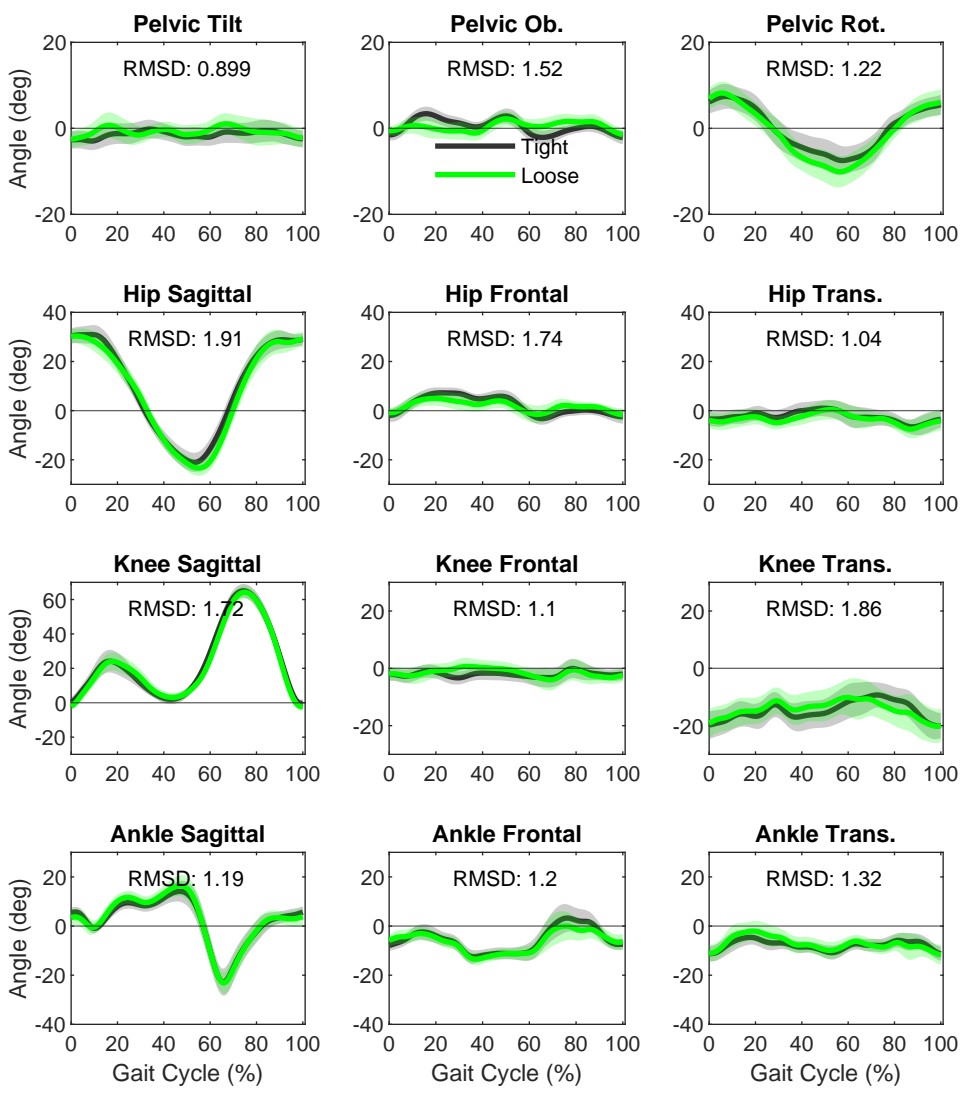

**Figure 4  Session one means and standard deviations (shaded) between the two types of clothing and root mean square difference (RMSD) between the clothing conditions is inset for each.** The green line represents loose clothing, and black line represent tight clothing.

**Table 4  Shank and thigh segment length differences between sessions 1 and 2 for tight and loose clothing.**

|  | Loose | | Tight | |
| --- | --- | --- | --- | --- |
|  | Shank | Thigh | Shank | Thigh |
| Inter-session difference (mm) | | | | |
| Mean | 10 | 9 | 3 | 29 |
| SD | 10 | 6 | 3 | 15 |
| Difference as % segment length (%) | | | | |
| Mean | 2 | 2 | 1 | 7 |
| SD | 3 | 2 | 1 | 4 |

of these magnitudes are "acceptable" and "reasonable" when compared to the suggested thresholds of variation expected for marker-based motion capture (*McGinley et al., 2009*). Variability was also below the 5° intra-tester threshold of CMAS (*Stewart et al., 2023*). Multiple independent studies now show encouraging inter-trial and inter-session reliability using markerless gait analysis. It is important to not only consider summarized variations but also the key features of gait kinematic curves beyond maxima, minima and ranges of motion (*Riazati et al., 2022*).

As no markers are placed on participants, and kinematics are determined algorithmically, inter-assessor variation does not apply to markerless analysis. This is significant as inter-assessor variation is generally greater than inter-trial and inter-session variation (*Schwartz, Trost & Wervey, 2004*; *Manca et al., 2010*). As CMAS thresholds are 5−10° for inter-assessor variation, different criteria and quality assurance should instead be considered for markerless protocols. To extend this to practical contexts, gait labs with multiple assessors may wish to consider markerless gait analysis as a way to avoid the increase in inter-assessor variation that a second gait analysis provides. Firstly though, additional studies considering clinical populations and children (*e.g.*, *McGuirk et al., 2022*; *Wren, Isakov & Rethlefsen, 2023*; *Outerleys et al., 2024*) are required. Differences in inter-laboratory variation see (*Kaufman et al., 2016*) would also likely be reduced when using the same markerless software, but this is yet to be determined empirically.

As this is the first time the reliability of markerless pelvis kinematics have been evaluated, pelvis variation warrants specific consideration. Pelvis kinematic variations were generally smaller than the other joints, which is consistent with marker-based studies (*Schwartz, Trost & Wervey, 2004*; *Manca et al., 2010*; *Kaufman et al., 2016*). In markerless pose estimation, low pelvis variation could be because the pelvis is modelled as a six-degree of freedom segment. Therefore, the pelvis is independent of pose requirements of other segments in the inverse-kinematic chain. It could also be the case that pelvis landmarks are more consistently identified algorithmically but this is not known. With respect to specific gait features, the tight clothing condition appeared to most appropriately represent the obliquity in the frontal plane whereas in the loose clothing condition, the frontal plane of the pelvis remained relatively neutral. Therefore, it remains to be seen if pathological movement of the pelvis can be described appropriately in loose clothing if a greater reliance

is placed on algorithmic determination of pelvis kinematics, which may or may not be included in the model training.

The average variation ratios of ∼1.5° for all joints and planes was notably higher than reported by *Kanko et al. (2021a)* of ∼1.1° . This is due to the smaller inter-trial variation seen in this study, which when considered against the additional ∼1° inter-session variation, leads to a larger overall ratio. Therefore, a second session increases the variation in joint kinematics by ∼50% irrespective of whether tight or loose clothing is worn. Nevertheless, this is still substantially smaller than variation ratios when a second assessor is involved (*Schwartz, Trost & Wervey, 2004*). Reductions in inter-trial (intrinsic) variation could be caused by a newer software version used in this study, this is discussed below.

The average RMSD found between clothing conditions was smaller in this study than what was previously reported by *Keller et al. (2022)* 1.4° *vs* 2.6°. The choice of tight *versus* loose clothing did not affect the inter-trial or inter-session variation. The higher variation in the transverse plane remains a consistent result with previous studies but lower variation overall means that these are now comparable with the sagittal plane in other studies (*Kanko et al., 2021a*; *Keller et al., 2022*; *Riazati et al., 2022*). Higher transverse plane variation (hip and knee) is also commonly reported in marker-based motion analysis (*Kanko et al., 2021b*; *Riazati et al., 2022*). The lack of difference between clothing conditions could be due to the deep learning algorithm being trained on publicly available images in a variety of clothing (*Kanko et al., 2021a*). The obvious benefits of wearing tight clothing when placing skin-mounted markers on the body does not seem to contribute to any substantial reduction in variation when compared to wearing loose fitting clothing. Tight clothing may in fact be more sensitive to where the shorts end or the top overlaps as these clothes are typically also dark and featureless. These issues may have contributed to the thigh segment length showing greater inter-session differences.

Inter-session-clothing variation was calculated to quantify the variation across sessions when participants wear different clothes (tight or loose). As it was calculated as a combination of the variation in both clothing conditions, an average variation of 2.35° approaches the most rigorous thresholds for marker-based data (*McGinley et al., 2009*). This is somewhat remarkable given it is an accumulation of inter-trial, inter-session and inter-clothing variations. It would therefore seem unnecessary for participants to change from everyday clothing into tighter fitting clothing for the purposes of markerless gait analysis. However, it should be noted that more challenging clothing (*e.g.*, a Jubah/thobe, long skirts, dresses) are untested to date yet Theia3D appears to cope well with additional accessories such as backpacks (*Coll et al., 2024*).

Inter-trial and inter-session variation was lower than in previous studies. As Theia3D software provides pose data based on identified features, any updates to the software may result in altered kinematics between versions. Previous reliability studies have used earlier version of the software (*Kanko et al., 2021a*)-unknown; (*Keller et al., 2022*)-v.2021.1.0.1450; (*Riazati et al., 2022*)-v.2021.2.0.1675. This current study used v2022.1.0.2309 which has resulted in lower inter-trial and inter-session variation and smaller differences between clothing conditions. Although we can only speculate that this could be due to version updates, it is seemingly important to report versions used to monitor the change in gait

kinematics as the software updates. For example, in March 2023 an update to the Theia software was described in a Theia blog (*Brown, 2023*, https://www.blog.theiamarkerless.ca/) which, amongst other changes, increased the number of tracked landmarks compared to the earlier versions and may continue to reduce variation. Although it is not thought possible to reduce intrinsic variation of marker based data (*Schwartz, Trost & Wervey, 2004*) this could be possible through algorithmic updates to the inverse-kinematic modelling. Careful monitoring of updates should also take place to ensure reductions in variation do not lead to increased centralization of data or the redistribution of variation into other planes.

Limitations of this study are that there was some variation in clothing conditions as some participants wore loose fitting shorts and t-shirts as their choice of regular wear, and, therefore, had exposed knees and elbows, and there was variation in the contrasting nature of the clothing worn. Although specific colours were not specified, colour contrast and contrast to the video background may be important to identify the anatomical features (*Kanko et al., 2021a*). Secondly, we considered right side gait data only. Thirdly these results are specific to the software version used for analysis. Since our analysis, further updates to software have been released. Finally, the walking speed between sessions and clothing conditions was not controlled in this study which may cause some increases in variation in the gait kinematic results.

## CONCLUSIONS

Variation in markerless motion capture data was within suggested thresholds of reliability for healthy adults. Specific consideration of the pelvis within this study found variation to be smaller than that of the hip, knee and ankle. Inter-session-clothing variation was <3.5°, which suggests that the choice of clothing across multiple sessions does not substantially increase variation substantially. Lower variation than in previous studies may be a result of continued algorithm development. As markerless technologies continue to demonstrate their reliability for healthy adult gait analysis, extending these studies to other populations and clinical context is now required.

## ACKNOWLEDGEMENTS

We would like to extend our gratitude to Karl Gibbon and Jiaming Xu for their support during data collection and assistance in tackling technical challenges in the preparation of this project. Your contributions and support were much appreciated.

### Funding

This work was funded by the Malaysian government agency, Majlis Amanah Rakyat (MARA), reference number: 330408371069. The funders had no role in study design, data collection and analysis, decision to publish, or preparation of the manuscript.

### Grant Disclosures

The following grant information was disclosed by the authors:
Majlis Amanah Rakyat (MARA): 330408371069.

### Competing Interests

Mark A. Robinson is an Academic Editor for PeerJ. Rich J. Foster is currently the Principal Investigator for a project based at Liverpool John Moores University, funded by Theia Markerless, Inc. No personnel or resources from Theia Markerless Inc. were involved in the generation or interpretation of the current data or manuscript. No other authors have conflicts to disclose.

### Author Contributions

- Sylvia Augustine conceived and designed the experiments, performed the experiments, analyzed the data, prepared figures and/or tables, authored or reviewed drafts of the article, and approved the final draft.
- Richard Foster conceived and designed the experiments, analyzed the data, authored or reviewed drafts of the article, and approved the final draft.
- Gabor Barton conceived and designed the experiments, authored or reviewed drafts of the article, and approved the final draft.
- Mark J. Lake conceived and designed the experiments, authored or reviewed drafts of the article, and approved the final draft.
- Raihana Sharir conceived and designed the experiments, performed the experiments, analyzed the data, authored or reviewed drafts of the article, and approved the final draft.
- Mark A. Robinson conceived and designed the experiments, analyzed the data, prepared figures and/or tables, authored or reviewed drafts of the article, and approved the final draft.

### Human Ethics

The following information was supplied relating to ethical approvals (i.e., approving body and any reference numbers):

Liverpool John Moores University Research Ethics Committee granted Ethical approval (UREC reference: 21/SPS/063).

### Data Availability

The raw data are available in the Supplementary Files.

### Supplemental Information

Supplemental information for this article can be found online at http://dx.doi.org/10.7717/peerj.18613#supplemental-information.

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
