# Peer review of "The inter-trial and inter-session reliability of Theia3D-derived markerless gait analysis in tight versus loose clothing"

_PeerJ, doi:10.7717/peerj.18613_

## Round 0.1 · original submission · Minor Revisions

Please pay attention to the reviewer comments. Including the comment regarding the Matlab code. Please update as per the suggestions provided.

·

Basic reporting

No comment

Experimental design

No comment

Validity of the findings

No comment

Additional comments

This study sought to examine variation in joint kinematics between trials, sessions, and clothing conditions using Theia3D, a popular commercial markerless motion analysis software. The manuscript is well-written and offers clear, useful application to biomechanists and practitioners. It fills a nice gap in the literature in that it considers clothing within the trial × session design. I commend the authors for this. Whilst this study is publishable and appropriate for this Journal, I do have several questions for authors which they address:

The findings of this study are only applicable to users of Theia3D, not other markerless motion analysis systems (e.g., OpenPose, etc.) As such, I wonder if the authors might consider including this important fact in the title of the manuscript, for example: “The inter-trial and inter-session reliability of markerless gait analysis in tight versus loose clothing using Theia3D”. I think this point (lack of transferability to other markerless systems) should be addressed in the study limitations section of the discussion.

I also think a clearer justification for comparing loose and tight clothing is needed in the introduction, particularly in the paragraph in line 114-124. Why is the inclusion of loose clothing important, especially in clinical populations? This is also briefly discussed in the discussion but could be strengthened to show a real benefit of this study. Additionally, the opening paragraph of the introduction (lines 65-71) could further solidify this point, as marker-based motion capture requires participants to wear tight-fitting clothing for obvious reasons.

Can the authors speculate why pelvis kinematics are generally ignored by these studies (lines 107-112)? I agree they are important, and the authors have justified why we should, but have not offered any reasons for why we have/do not. Could it be because it is an “absolute” angle as opposed to a “relative” angle, thus is more likely to be affected by the model development? Although the pelvis is tracked in 6DOF by Theia3D, it is possible that constraints are placed on its ability to rotate relative to the static pose. This is applicable to the paragraph exploring this in the discussion too (lines 243-255).


There are also a few specific, yet minor, comments:

Lines 89-90: SIMI Motion’s markerless motion analysis system is called “SIMI Shape 3D”: http://www.simi.com/en/products/movement-analysis/markerless-motion-capture.html

Lines 94-97: More explanation is required regarding the reasons for considering inter-session reliability of an automated markerless motion capture system. What factors mean that this should be considered? For example, calibration, camera locations, variable lighting (e.g., if the lab has windows).

Protocol or Data Analysis: Was a static trial conducted for each clothing condition and on each session? Or did the authors use the statics option where an “average” pose is taken from all trials. Please provide some detail on this.

Line 141: Height of volume?

Lines 148-152: Were the cameras evenly distributed around the capture volume?

Line 198: You are missing a full stop after “<1°”.

Lines 266-267: Should the numerical values go in parentheses?

Figure 3 and 4: Nice figures. I notice that the hip angle seems to be more extended around toe-off in loose clothing. Do the authors think this is clothing-related, or potentially because of any confounding factor like gait speed? Perhaps this can be discussed somewhere (e.g., a limitation here is that speed was not controlled between trials, sessions, or clothing conditions).

·

Basic reporting

General statement: This article compared inter-trial and inter-day variability across different clothing conditions (i.e., tight vs loose clothing) for the markerless motion capture system Theia3D. Acceptable reliability results were found for all lower limb angles and pelvis, leading the authors to conclude that Thiea3D is a reliable method to use in data collection sessions regardless of clothing fit. Along with this document, I have supplied an annotated version of the manuscript where I provide 80+ comments that focus on style suggestions, correct grammatical errors, and general discussion. I expect the authors to carefully consider the comments in the annotated manuscript, but there is no need to respond to them; a response is only required for the comments here in the formal reviewer responses.

1. Professional English is largely used throughout. This is addressed in the annotated manuscript.
2. The introduction is well organized, clearly defines terms, and puts the research within the context of previous literature. However, the authors take liberties in using concepts and terminology that a non-expert may not know. You should never expect a reader to search out other literature to comprehend your work. In that regard, I advise the authors to expand on what gait indices are/why they are important for surgical interventions. Lines 108-112.
3. Raw data are shared. However, the code provided is not up to standards. The Matlab code is not built to work with the provided csv files. It is designed to load in a mat file that is not shared. To that end, the code is poorly commented and provides no description of what the code does or what each csv file is. Variable names are not interpretable, nor are the acronyms used defined. When you are developing code for yourself, it is acceptable to use short forms and acronyms; however, when you share code with others, it should be a standalone document that can be interpreted by anyone at any skill level. You achieve this universal interpretation by creating detailed comments and defining variables with full names. If an acronym is necessary, define it. Please provide an updated version by taking into account this feedback.
4. Please correct Figure 2 such that the legend does not cut off any of the data in rows 2 and 3.

Experimental design

The article is within the scope of the journal.
The research question is well-defined and is meaningful in a clinical and academic context.
Methods are largely described in detail. Questions will address this.

1. Please provide the reader with the average and standard deviation for the time between sessions. It states within 14 days, but more information about the time frame would be nice.
2. Please clarify the clothing choice between sessions. It is unclear to me if the participants wore the same clothes on two different days or wore different outfits. I.e., did participants wear up to four different outfits or the same two (tight, loose) between days?
3. Please provide your motivations for choosing one gait cycle per trial and only the right side of the body. It is unclear from the manuscript why this decision was made. Where in the walking pattern was this walk taken from? Steady-state walking? How did your team control for selecting a similar gait cycle between trials? Why the right side and not the left? This information will help the reader understand the methodological decisions and engage more deeply with your research.
4. I left a comment in the manuscript, but I would like you to clarify lines 177-182 here. The authors repeatedly state, “five trials,” but from my understanding, 10 trials were collected per session. If you mean five trials per clothing condition, I would like that to be clearly stated.

Validity of the findings

I believe that the conclusion is well stated. The authors relate their findings using pre-defined thresholds that they explained in the introduction and referenced throughout.
Data are provided, but as stated in the above sections, the code that goes with them is ill-equipped to assess the data.

1. Throughout the results, I would like clarification around the statements “considered separately.” It is not clear to me what separate things are being considered in the different contexts.
2. The authors report joint angle variability; however, I am curious about segment lengths. All participants are adults, and we can assume they did not undergo skeletal changes within 2 weeks. Can the authors comment on whether Theia calculated the same segment length for the tibia and femur between and within sessions?

Additional comments

The authors mention on line 285 that the effects of backpacks have not been assessed. Unfortunately, our manuscript is currently under review; however, a master's student in our group has assessed the effects of bulky military equipment and backpacks on Theia3D's ability to track the lower limbs and their effects on kinematics/kinetics. If the authors are interested, her master's thesis is entitled Validation of Markerless Motion Capture for the Assessment of Soldier Movement Patterns Under Varying Body-Borne

---

## Round 0.2 · accepted · Accept

Thank you for your thoughtful revision. I am satisfied that you met the intent of the reviewers and appeased their concerns. This paper reads well and I believe adds to the current literature.

I note one typo on line 183 "Cameras were position closer " should be "Cameras were positioned closer ", please ensure this is fixed up during proofing.